# Clinical Course, Outcomes, and Risk Factors of Myocarditis and Pericarditis Following Administration of mRNA-1273 Vaccination: A Protocol for a Federated Real-World Evidence Vaccine Safety Study Using Data from Five European Data Sources

**DOI:** 10.3390/vaccines13070755

**Published:** 2025-07-16

**Authors:** Laura C. Zwiers, Diederick E. Grobbee, Rob Schneijdenberg, Corine Baljé, Samantha St. Laurent, Daina B. Esposito, Lei Zhu, Veronica V. Urdaneta, Magalie Emilebacker, Daniel Weibel, Felipe Villalobos, Carlo Alberto Bissacco, Arantxa Urchueguía Fornes, Juan José Carreras-Martínez, Anteneh A. Desalegn, Angela Lupattelli, Lei Wang, Jannik Wheler, Vera Ehrenstein, Denise Morris, Catherine Fry, Marjolein Jansen, Brianna M. Goodale, David S. Y. Ong

**Affiliations:** 1Julius Global Health, Julius Center for Health Sciences and Primary Care, University Medical Center Utrecht, Utrecht University, 3584 CX Utrecht, The Netherlands; d.e.grobbee@umcutrecht.nl (D.E.G.); davidsyong@gmail.com (D.S.Y.O.); 2Julius Clinical, 3703 CD Zeist, The Netherlands; 3Clin-Q B.V., 9723 DJ Groningen, The Netherlands; 4ModernaTX, Inc., Cambridge, MA 02142, USA; 5Vaccine Monitoring Collaboration for Europe (VAC4EU), 1000 Brussels, Belgium; weibel@vac4eu.org; 6Fundació Institut Universitari per a la Recerca a l’Atenció Primària de Salut Jordi Gol i Gurina (IDIAPJGol), 08029 Barcelona, Spaincabissacco@idiapjgol.info (C.A.B.); 7Vaccine Research Department, Foundation for the Promotion of Health and Biomedical Research in the Valencian Region (FISABIO—Public Health), 46020 Valencia, Spain; 8Centro de Investigación Biomédica en Red de Epidemiología y Salud Pública, Instituto de Salud Carlos III, 28029 Madrid, Spain; 9Pharmacoepidemiology and Drug Safety Research Group, Department of Pharmacy, Faculty of Mathematics and Natural Sciences, University of Oslo, 0313 Oslo, Norway; a.a.desalegn@farmasi.uio.no (A.A.D.);; 10Department of Clinical Epidemiology, Aarhus University Hospital and Aarhus University, 8200 Aarhus, Denmark; lei.wang@clin.au.dk (L.W.); janwhe@clin.au.dk (J.W.); ve@clin.au.dk (V.E.); 11Drug Safety Research Unit, Southampton SO31 1AA, UK; denise.morris@dsru.org (D.M.); catherine.fry@dsru.org (C.F.); 12School of Pharmacy and Biomedical Sciences, University of Portsmouth, Portsmouth PO1 2UP, UK; 13Department of Medical Microbiology and Infection Control, Franciscus Gasthuis & Vlietland, 3045 PM Rotterdam, The Netherlands

**Keywords:** myocarditis, pericarditis, vaccination, COVID-19 vaccines, safety

## Abstract

**Background**: Myocarditis and pericarditis are recognised risks following COVID-19 vaccination, including the mRNA-1273 vaccine. Most cases occur shortly following the second dose of this vaccine, and incidence is highest among young males. However, little is known about risk factors beyond age and sex and about the longer-term clinical course. This study aims to identify possible risk factors for myocarditis and pericarditis following mRNA-1273 vaccination, to characterise the clinical course of myocarditis and pericarditis, both associated with mRNA-1273 vaccination and not associated with vaccination, and to identify risk factors for severe outcomes (i.e., cardiac or thromboembolic complications, severe hospital outcomes, all-cause hospital readmission, and death). **Methods**: This study is being conducted within the Vaccine Monitoring Collaboration for Europe (VAC4EU) association using routinely collected healthcare data from five data sources from four European countries (Denmark, Norway, Spain, and the United Kingdom). The study is being performed using a common data model, and all analyses are performed separately in each data source in a federated manner following a common protocol. A case–cohort analysis set is identified within each data source for identifying potential risk factors for myocarditis and pericarditis following mRNA-1273 vaccination using logistic regression analysis. The clinical course of myocarditis and pericarditis is being assessed using a cohort study design and describes all cases (i.e., cases associated with mRNA-1273 and unexposed cases). Cox regression analysis is applied to assess the associations between risk factors and several follow-up outcomes. **Conclusions**: This protocol describes the study methodology of an international collaborative initiative with the aim of assessing the risk factors and clinical course of myocarditis and pericarditis following mRNA-1273 vaccination using a federated network of five European data sources.

## 1. Introduction

The severe acute respiratory syndrome coronavirus 2 (SARS-CoV-2) has led to a global pandemic of the coronavirus disease 2019 (COVID-19). During this pandemic, vaccines were rapidly developed and approved for administration. Since the administration of the first dose of a COVID-19 vaccine in 2020, over 13.5 billion COVID-19 vaccine doses have been administered worldwide as of April 2024 [1]. Several adverse events were observed following vaccination, including rare cases of myocarditis and pericarditis. These events were identified as risks following the COVID-19 mRNA vaccines from Pfizer and Moderna (BNT162b2 and mRNA-1273) [2,3,4], but also following the ChAdOx1 and NVX-CoV2373 vaccines [5,6].

The incidence of myocarditis and pericarditis following COVID-19 mRNA vaccination is highest in adolescent and young adult males, and symptoms generally occur around a median of three days after the second vaccine dose [7,8,9,10,11]. Among individuals aged 12 to 39, the combined incidence of myocarditis and pericarditis following mRNA vaccination has been estimated to be 12.6 cases per one million second doses of a vaccine [12]. A large European multinational cohort study found that the incidence rate ratios for myocarditis in individuals aged under 30 were 3.3 and 7.8 after the first and second BNT162b2 vaccine doses, and 3.5 and 6.1 after the first and second doses of the mRNA-1273 vaccine [10]. It has also been suggested that there are no differences in incidence between heterologous and homologous dosing schedules in the primary vaccination series, and that the incidence of myocarditis and pericarditis is lower after longer intervals (e.g., over 30 days between vaccine doses) [11]. Still, little is known about risk factors apart from age, sex, and dosing schedules.

To evaluate the risks and benefits of vaccination, it is important to describe the clinical features (e.g., cardiac complications, thromboembolic complications, hospital readmissions, intensive care unit (ICU) admissions, and deaths) of cases of myocarditis and pericarditis following vaccination and compare them to cases that occur without exposure. Existing research suggests that individuals experiencing myocarditis or pericarditis after vaccination experience less severe outcomes compared to those with myocarditis or pericarditis not related to vaccination [13,14,15]. However, more research is needed to fully understand how the clinical course compares between mRNA-1273-exposed and unexposed cases, and which factors are associated with unfavourable outcomes.

## 2. Objectives

This study investigates whether factors other than age and sex contribute to the risk of developing myocarditis or pericarditis following mRNA-1273 vaccination. Moreover, it describes the clinical course of myocarditis and pericarditis of varying aetiologies. Specifically, the primary aims are

To identify possible risk factors for myocarditis and pericarditis following mRNA-1273 vaccination, including demographic characteristics, medical history, and vaccination characteristics.To characterise the clinical course of myocarditis and pericarditis of varying origin, including myocarditis and pericarditis associated with mRNA-1273 vaccination, and myocarditis or pericarditis not associated with vaccinations targeting SARS-CoV-2, and to identify prognostic factors of the course of myocarditis and pericarditis.

In addition, this study aims to provide additional insight into the differences in clinical course between outcomes following vaccination and outcomes not following vaccination, as well as the risk factors of more severe outcomes. For this part, there are two secondary objectives:To identify whether there are differences in the clinical course and risk factors between myocarditis and pericarditis associated with mRNA-1273 vaccination, and myocarditis and pericarditis not associated with vaccinations targeting SARS-CoV-2.If severe cases or cases with sequelae are identified, to identify risk factors for severe myocarditis and pericarditis associated with mRNA-1273 vaccination.

The study has been registered in the HMA-EMA Catalogues of real-world data sources and studies, and a full detailed protocol can be accessed through the catalogues (EU PAS number is EUPAS105009).

## 3. Methods

### 3.1. Study Design

To identify potential risk factors for the development of myocarditis and pericarditis following mRNA-1273 vaccination (i.e., primary objective 1), an mRNA-1273-exposed case–cohort analysis is being performed. Within the study population, a base cohort of individuals who received mRNA-1273 during the study period is identified. From this base cohort, everyone who developed myocarditis/pericarditis within 30 days following vaccination is included as cases, as well as a random sub-cohort of controls. Cases are explicitly selected within a 30-day time window following vaccination, as the previous literature suggests that the majority of cases occur within this time window, and most of the existing literature has investigated similar or even shorter exposure windows [8,10]. Random sampling of controls is performed based on the month of vaccination receipt, with four controls per case [16].

To characterise the clinical course of myocarditis and pericarditis (i.e., primary objective 2), a cohort of myocarditis/pericarditis cases is identified. This cohort includes individuals diagnosed with myocarditis or pericarditis within 30 days after mRNA-1273 vaccination (exposed) and those who did not receive any COVID-19 vaccine in the 30 days before myocarditis or pericarditis onset (further referred to as unexposed cases). Within this design, cases following vaccination are compared to a sample of unexposed cases on both short-term (30 days) and longer-term (90 days, 6 months, and 12 months) clinical outcomes following myocarditis or pericarditis onset, with the index date being the date at which one of these two events was recorded. A graphical illustration of both study designs for myocarditis as the outcome is shown in Figure 1, which also applies to pericarditis as a separate analysis.

### 3.2. Data Sources

This study is being performed using secondary health data collected in four European countries: Denmark, Norway, Spain, and the United Kingdom (UK). Data come from five different data sources and cannot leave local servers. Therefore, all analyses are performed locally in a federated manner. To access the data, we work with data experts and access partners (DEAPs) from the Vaccine Monitoring Collaboration for Europe (VAC4EU) association’s network. The DEAPs are able to access the data from various registries and data custodians, transform the data, and perform analyses on these data. The data comprise electronic, routinely collected healthcare data, and the databases cover the countries’ or regions’ underlying population fully or partially and provide representative samples of the underlying populations. Hence, this study is able to provide real-world insights on the risk factors and clinical course of myocarditis and pericarditis following mRNA-1273 vaccination. Table 1 provides a description of the different data sources.

### 3.3. Data Management

A common study protocol, common statistical analysis plan, and common data model (CDM) are used for this study. All analyses are conducted in a federated manner such that personal-level data never leave their local servers at each DEAP. The DEAPs extract the data required for the study and transform these data into the ConcePTION CDM [28] in an extraction-transformation-load (ETL) procedure. To verify the correctness of the data, quality checks are performed following a common programme provided by VAC4EU in collaboration with VAC4EU member “University Medical Center Utrecht, NL” [29]. Results of these quality checks are assessed by the study team before performing any study-specific analyses.

Upon completion of the data quality checks, study-specific analyses are performed through study scripts created in R. These scripts undergo quality control within the study team before being deployed to DEAPs. DEAPs run the scripts locally and upload aggregated, privacy-preserving results to the digital research environment (DRE), which is a cloud-based, globally available research environment were aggregated data are stored and organised securely under the responsibility of the University Medical Center Utrecht. The DRE applies double authentication where researchers can collaborate using data that are stored and organised securely. The study team can download results for further processing and interpretation after they have been uploaded to the DRE.

### 3.4. Study Period

Recipients of the mRNA-1273 vaccine and cases of myocarditis and pericarditis are identified in the respective databases retrospectively. The study start date is 6 January 2021, which was the date of earliest approval of the vaccine in Europe [30]. Since different DEAPs receive updated data at different frequencies and need time to transform and clean the data, the latest date of follow-up varies across DEAPS. The end of follow-up for each DEAP is outlined in Table 1.

### 3.5. Inclusion and Exclusion Criteria

In the case–cohort analysis, individuals are included if they (1) received at least one dose of mRNA-1273 during the study period, (2) have been enrolled in the applicable database for at least one year prior to vaccination to allow for ascertainment of risk factors, and (3) did not experience myocarditis or pericarditis in the six months prior to vaccination. Cases are included in the main case–cohort analysis if they meet adjudication criteria according to the Brighton Collaboration Case Definition [31].

In the cohort analysis, individuals are included if they (1) have been in the database for at least one year prior to the onset of myocarditis or pericarditis, and (2) meet adjudication criteria as described above. Persons with no record of mRNA-1273 vaccination, but with a record of another vaccine targeting SARS-CoV-2 in the 30 days prior to myocarditis/pericarditis onset, are excluded to ensure that a comparison can be made specifically between post-mRNA-1273 cases and cases not associated with any COVID-19 vaccination. In the main analyses, all cases following mRNA-1273 are included and validated, as well as a matched sample of unexposed cases. Exposed and unexposed cases are matched 1:1 based on age and sex, in order to limit the number of cases for which medical chart review needs to be performed.

### 3.6. Case Adjudication

Cases of myocarditis and pericarditis are initially identified in the databases through diagnostic codes (e.g., ICD-10 codes), but this identification may suffer from outcome misclassification [32]. There is uncertainty on the expected magnitude of misclassification and its potential impact on study results [32,33], but the Federal Drug Administration’s (FDA) draft guidance on studies employing real-world data states that validation of outcome variables is expected to minimise outcome misclassification [34]. Therefore, myocarditis and pericarditis cases are adjudicated by case validation through an electronic data capture form (e-DCF) in REDCap (the electronic data capturing software used; https://projectredcap.org/software/ (accessed on 13 July 2025)), based on the Brighton Collaboration Case Definition [31] in this study. Cases are only included in the main analyses if they have been adjudicated and meet the criteria for being a definite, probable, or possible case of myocarditis or pericarditis. Medically trained adjudicators at each DEAP have been instructed on the use of the form before performing the validation. Adjudication of cases is being performed through medical chart review and/or review of patient profiles generated by compiling cumulative chronologically ordered records of all relevant available electronic information in a data source. Results from the adjudication are made available to the study team in an aggregated, privacy-preserving manner to allow for further use of these results in the study. As feasible, adjudicators are blinded to vaccine exposure, such that differential misclassification is minimised.

### 3.7. Outcomes and Covariables

#### 3.7.1. Case–Cohort Analysis

In the mRNA-1273-exposed case–cohort analysis, the outcome of interest is myocarditis or pericarditis within 30 days after mRNA-1273 vaccination. The onset of myocarditis/pericarditis is defined as the record of one of these outcomes, identified through medical codes in the respective databases (e.g., ICD10 diagnosis codes, coding systems for each DEAP are listed in Table 1), with cases undergoing adjudication after having been identified. Identification through disease codes is performed using code lists created and clinically reviewed by the VAC4EU code review task force. Through using these code lists, the identification of recorded cases is highly accurate, but cases not recorded in the data sources remain undetectable. The incidence of reported myocarditis and pericarditis following mRNA-1273 receipt will be compared to preexisting data to assess the likelihood of over- or underreporting.

Due to the exploratory nature of this study, there are many covariables of interest. The potential risk factors of interest include demographic characteristics (age and sex), selected comorbidities, Charlson Comorbidity Index (CCI) score [35], history of relevant medical treatments, history of SARS-CoV-2 infection, history of vaccinations targeting SARS-CoV-2 and other vaccinations, healthcare utilisation in the year prior to myocarditis/pericarditis onset, lifestyle factors, and socioeconomic status. For both mRNA-1273 vaccination and other vaccinations targeting SARS-CoV-2, it is possible to ascertain the vaccine manufacturer, dosing order, schedule, and interval, if the vaccine was administered and/or recorded within the health system that the respective database or registry covers.

The exact variables may differ per data source due to data availability. An overview of the exact determinants of interest and how they are measured can be found in Appendix A, lists of all comorbidities, medical treatments, and vaccinations of interest are also available in Appendix A.

#### 3.7.2. Cohort Analysis

There are several outcomes of interest in the myocarditis/pericarditis cohort analysis. Specifically, the characterisation of the clinical course of myocarditis/pericarditis is performed by assessing the following clinical outcomes at various follow-up windows after the onset of myocarditis or pericarditis: cardiac complications (including acute coronary syndrome, acute myocardial infarction, heart failure, atrial fibrillation/flutter, ventricular arrhythmias, and cardiac arrest); thromboembolic complications (pulmonary embolism or deep venous thrombosis, stroke outcomes, and peripheral arterial embolism); hospital readmission; severe hospital outcomes (hospital readmission, ICU admission, and death, as a composite); and death. The follow-up windows are 0–30 days, 0–90 days, 0–6 months, and 0–12 months. Additionally, the acute phase of care is described with respect to the highest level of care and length of stay in the hospital. All outcomes are identified through medical codes or hospital data recordings in the respective databases.

In the cohort analysis, the covariables of interest are similar to those in the case–cohort analysis. Additionally, recent exposure to mRNA-1273 is assessed as a covariable in this analysis. Within this analysis, an individual is considered recently exposed if they received mRNA-1273 within 30 days prior to myocarditis/pericarditis onset.

### 3.8. Statistical Analysis

#### 3.8.1. Case–Cohort Analysis

To limit biases, the sampling of controls in the case–cohort analysis is being performed 1000 times, such that all analyses are performed for each of the 1000 runs separately and results are pooled across these runs. For the descriptive analysis, all cases and the average of the 1000 sampled control cohorts are described in terms of demographic characteristics, medical history, lifestyle factors, socioeconomic variables, and vaccination characteristics, including the number of doses received. Additionally, descriptive analyses include an overview of the number of individuals included in each data source and the number who received the mRNA-1273 vaccine, such that the proportion of the population who received this vaccine can be deducted. Subsequently, a logistic regression methodology is applied within each of the 1000 runs, with the outcome of interest being myocarditis/pericarditis within 30 days following receipt of mRNA-1273, and the variables of interest as listed in the previous section. A variable selection procedure where candidate covariables are first identified using univariable testing, followed by forward selection in the multivariable model, is performed. This is further detailed in Appendix A. Final estimates, odds ratios, and corresponding 95% confidence intervals (CIs) will be presented for these variables. Significant associations are defined as those for which the 95% confidence interval does not contain the value 1.

#### 3.8.2. Cohort Analysis

Descriptive statistics are presented separately for cases following mRNA-1273 vaccination (exposed) and cases not following vaccination (unexposed) to characterise and identify differences in the clinical course (Primary objective 2) and to identify differences in risk factors (Secondary objective 1). Myocarditis/pericarditis cases are separately described in terms of demographic characteristics, medical history, lifestyle factors, socioeconomic variables, and vaccination characteristics, including the number of received doses, and are compared based on these variables. Descriptive statistics are also presented separately for severe cases (Secondary objective 2). Severe myocarditis is defined as the presence of myocarditis plus the presence of at least one of the following: acute heart failure, cardiogenic shock, atrial fibrillation/flutter, or ventricular arrhythmias/cardiac arrest. Severe pericarditis is defined as the presence of pericarditis plus the presence of fever, cardiac tamponade, and/or the performance of pericardiocentesis [36].

To identify risk factors for the outcomes of interest, Cox proportional hazards models on the combined exposed and unexposed cases are used. Cox regressions are performed separately for each of the follow-up windows. Variable selection is performed to identify candidate risk factors for each outcome, and coefficients, hazard ratios, and corresponding 95% CIs are presented for the final model. All analyses across both study designs are performed using analytical R scripts, which are centrally created and run locally at the DEAPs’ servers.

#### 3.8.3. Subgroup Analyses

Subgroup analyses are performed if the sample size allows. If insufficient, analyses may be limited to descriptive statistics.

In the case–cohort set, subgroup analyses based on age and sex are planned. Analyses are being performed separately for young children (<12 years), adolescents (12–17 years), young adults (18–24 years), adults (≥25 years), males overall, females overall, and males aged between 18 and 30 (inclusive) years [8,37,38].

In the cohort set, subgroup analyses are conducted by exposure status (recent exposure, past exposure, or never exposed, with recent exposure being mRNA-1273 exposure within 30 days prior to myocarditis/pericarditis onset, and past exposure being mRNA-1273 exposure more than 30 days prior to onset), SARS-CoV-2 infection status (previously infected or never infected), index year, and vaccine variant, if available in the respective database.

#### 3.8.4. Sensitivity Analyses

Sensitivity analyses focus on the robustness of results. A first set of sensitivity analyses to be performed entails including cases that could not be adjudicated, or did not meet the criteria for definitive, probable, or possible cases according to the Brighton Collaboration Case Definition of myocarditis or pericarditis. Further sensitivity analyses are performed in which the window for defining recent mRNA-1273 exposure is shortened from 30 days to 7 and 14 days. Finally, in the case–cohort analysis, an additional sensitivity analysis is performed in which individuals with a history of myocarditis/pericarditis more than six months prior to the index date are also excluded.

## 4. Discussion

This protocol was developed to investigate potential risk factors for myocarditis and pericarditis following mRNA-1273 vaccination and to characterise the clinical course of these conditions, including an assessment of risk factors for more severe clinical outcomes. To our knowledge, it is one of the first studies to investigate a broad range of potential risk factors beyond age and sex for myocarditis and pericarditis following vaccination in a real-world setting. A case–cohort and cohort study design were used in this study. The case–cohort analysis included mRNA-1273 recipients, while the cohort analysis included all myocarditis and pericarditis cases, both following mRNA-1273 vaccination and of other aetiology, from five databases spanning four European countries. The data accessed through the DEAPs from the VAC4EU association’s network provides full, or at least fully representative, population coverage of the regions that the respective DEAPs are located in, thereby minimising selection bias. Additionally, through the use of the ConcePTION CDM [28], a common protocol, common statistical analysis plan, and common scripts, data are analysed through a fully federated network in which individual-level data never leave their local servers. The study’s data management framework ensures privacy while enabling the possibility to study a diverse European population.

This is also one of the few studies to date with the specific aim of characterising the clinical course of myocarditis and pericarditis and assessing risk factors for the clinical course of these diseases in a broad, real-world population. Insights gained from this study on the clinical course may be valuable to both clinicians and patients, potentially contributing to shape future treatment pathways. Moreover, we hope the study methodology and the insights from this study will stimulate further investigation into the risk factors and clinical course of adverse events following vaccinations, including myocarditis and pericarditis. An additional strength of this study in comparison to other studies investigating the clinical course is the ability to investigate follow-up outcomes after myocarditis/pericarditis onset with minimal loss-to-follow-up due to the nature of the secondary health databases used.

There are several biases associated with the use of secondary data in this study. Firstly, some of the variables of interest, especially those related to lifestyle and socioeconomic status, are not available or comprehensively recorded in all databases. Those variables can, therefore, not be studied as potential risk factors in all countries. Moreover, the incidence of myocarditis/pericarditis following mRNA-1273 is quite low, with the combined incidence of myocarditis/pericarditis estimated to be 12.6 per one million second-dose mRNA vaccines targeting SARS-CoV-2 among individuals aged 12 to 39 [12]. For myocarditis alone, reporting rates were approximately 4.8 cases per million doses of mRNA vaccines in individuals aged 12 and older [39]. These low incidence rates of myocarditis and pericarditis may limit the number of individual covariables of interest that can be included in the statistical models. Furthermore, due to the inclusion criteria, this study may miss myocarditis or pericarditis cases that occur more than 30 days after vaccination, even if those cases could still be associated with the vaccination.

There is a potential for selection bias in this study. First, more severe cases of myocarditis and/or pericarditis may have been more likely to be recorded in the data sources, leading to their overrepresentation in the study samples. In contrast, milder cases may have gone unrecorded and, thus, were potentially missed. Moreover, the decision to receive the vaccine may have been influenced by underlying health status. Those with more pre-existing comorbidities may have been more likely to be vaccinated, introducing a potential selection bias in the base cohort of vaccinated individuals.

The sampling of unexposed cases to be performed in the cohort study may pose a limitation to the characterisation of those cases. As most myocarditis/pericarditis cases following vaccination are expected in young adults [7,8,37,38], age-matched unexposed myocarditis/pericarditis cases will also mainly comprise young adults. Unexposed cases in older individuals will, thus, be under-represented in these analyses. This limitation will be addressed through the sensitivity analyses, including all cases, regardless of whether validation was performed.

Finally, while this study provides an insight into the potential risk factors and clinical outcomes of myocarditis/pericarditis following mRNA-1273 vaccination, it does not provide comparisons with outcomes following other vaccinations targeting SARS-CoV-2. Similar research on other vaccines is, thus, encouraged, such that risk factors and clinical course can be compared and vaccination strategies potentially adapted.

## 5. Conclusions

This protocol describes the study design and methodology of a large international collaborative initiative aimed at assessing the risk factors and clinical course of myocarditis and pericarditis following mRNA-1273 vaccination. We anticipate that the presented methodology will facilitate the generation of valuable insights from real-world data.

## Figures and Tables

**Figure 1 vaccines-13-00755-f001:**
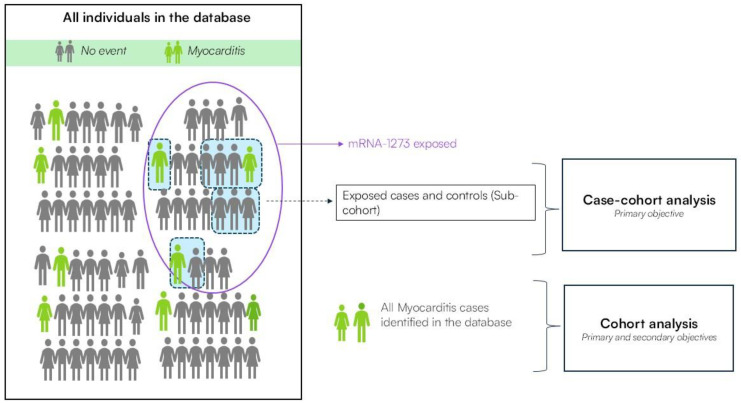
Graphical representation of both study designs.

**Table 1 vaccines-13-00755-t001:** Brief overview of characteristics of data sources used in this study.

Country	Denmark	Norway	Spain (Catalonia)	Spain (Valencia)	UK
DEAP name	Aarhus University	University of Oslo	IDIAP	FISABIO	DSRU
Data source name	Danish national registries [17,18,19]	Norwegian national registries [20,21,22,23,24]	SIDIAP [25]	VID [26]	CPRD Aurum [27]
Total population covered in data source *	5.8 million	5.5 million	5.8 million	5.0 million	16.2 ^#^ million
Population coverage	Entire Danish population	Entire Norwegian population	75% of the Catalan population	Almost entire (~96%) population of the Valencia region in Spain	Around 24.2% of the UK population ^#^
Date of latest follow-up	15 April 2023	31 December 2022	31 December 2023	31 December 2022	23 January 2024
Medical coding systems used	ICD10DA, ATC	ICD10, ICPC2, ICPC2B, ATC	ICD10CM, ICD10PCS, ATC	ICD9CM, ICD10CM, ICD10ES, ATC	medcodeid, prodcodeid

* Most recent estimates from the VAC4EU catalogue on data sources (accessed on 14 November 2024); ^#^ Percentage of the UK population coverage (current patients only, i.e., registered at currently contributing practices, excluding transferred out and deceased patients): 16,184,439 of 67,026,300 (24.15%); SIDIAP = Sistema d’Informació per al Desenvolupament de la Investigació en Atenció Primària; VID = Valencia Health System Integrated Database; DSRU = Drug Safety Research Unit; CPRD = Clinical Practice Research Datalink.

## Data Availability

The original contributions presented in this study are included in the article/Appendix A. Further inquiries can be directed to the corresponding author.

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
