# Peer review of "Clinical Course, Outcomes, and Risk Factors of Myocarditis and Pericarditis Following Administration of mRNA-1273 Vaccination: A Protocol for a Federated Real-World Evidence Vaccine Safety Study Using Data from Five European Data Sources"

_vaccines, 2025, doi:10.3390/vaccines13070755_

Round 1
Reviewer 1 Report
Comments and Suggestions for Authors
First of all I would like to congratulate you with regards to your paper . I do agree that I have not come across any other significant paper taking about the side effects of the COVID vaccine in greater depths other than talking about age and sex . I do agree there is potential of bias with regards to life style choices , preexisting medical conditions, type of medicines the patient is on or other vaccines being used to combat COVID -19 infections but the intent is good and the point which you were trying to make is aptly demonstrated In your work . Further more studies would be needed in depth to dwelve deeper . The statistical studies used in this study were robust . The only catch there could be multiple biases as spoken about earlier . Overall a good initiative and a good paper .
Author Response
Comments 1: First of all I would like to congratulate you with regards to your paper . I do agree that I have not come across any other significant paper taking about the side effects of the COVID vaccine in greater depths other than talking about age and sex . I do agree there is potential of bias with regards to life style choices , preexisting medical conditions, type of medicines the patient is on or other vaccines being used to combat COVID -19 infections but the intent is good and the point which you were trying to make is aptly demonstrated In your work . Further more studies would be needed in depth to dwelve deeper . The statistical studies used in this study were robust . The only catch there could be multiple biases as spoken about earlier . Overall a good initiative and a good paper .
Response 1:
We thank the reviewer for these kind words and the approval of our manuscript. Indeed, there are some potential biases to be considered with this study, which we have addressed in the discussion.
Reviewer 2 Report
Comments and Suggestions for Authors
Review:
The manuscript describes a protocol for studing the safety of the mRNA-1273 vaccine against SARS-CoV-19 regarding the incidence of myocarditis and pericarditis after the administration of this vaccine in five European countries. From a public health perspective, the protocol is important. Some suggestions could be indicated with the objective of improving the manuscript.
- Other SARS-CoV-2 vaccines have been associated with myocarditis and pericarditis, and they could be studied at the same time.
- Regarding the possibility of underdiagnosed myocarditis and pericarditis, and the consequent low reporting, how will it be addressed in each center?
- In order to a more precise knowledge, have been estimated the sensitivity and specificity of the reported myocarditis and pericarditis cases in each center?
- Considering the incidence of cases, an approximation of the power of the study to detect significant differences between potential risk factors and outcomes could be reported.
- An explanation of the methodology for the control of potential confounding factors could be indicated for a better understanding of the study.
- The protocol indicates that variables included in the multivariable models are based on statistical criteria without considering the potential confounding factors. This situation could be clarified.
- The authors mention some limitations of the study. However, a potential selection bias could occur, regarding the severity of the cases.
Author Response
Comments 1: Other SARS-CoV-2 vaccines have been associated with myocarditis and pericarditis, and they could be studied at the same time.
Response 1: We thank the reviewer for this suggestion. This study was registered as an EUPAS study and specifically designed to investigate the myocarditis/pericarditis safety signal following mRNA-1273 vaccination. Indeed, other vaccinations have also been associated with these outcomes, but those are beyond the scope of this study. We agree that more research into the risk of myocarditis and/or pericarditis following those other vaccines is still warranted.
Comments 2: Regarding the possibility of underdiagnosed myocarditis and pericarditis, and the consequent low reporting, how will it be addressed in each center?
Response 2: Unfortunately, we are not able to fully rule out underreporting of myocarditis and pericarditis. We have detailed information about how these outcomes are diagnosed and reported in each data source and work with extensively reviewed algorithms for identifying cases based on medical coding (line 227). By using these algorithms, we have minimized the potential for underreporting to the extent feasible in a real-world evidence study like ours.
Comments 3: In order to a more precise knowledge, have been estimated the sensitivity and specificity of the reported myocarditis and pericarditis cases in each center?
Response 3: The sensitivity and specificity of myocarditis and/or pericarditis diagnoses in each center have not been estimated previously, and we were not able to do this using the data available to us. This is because it is not possible to identify cases of myocarditis and/or pericarditis unless they are specifically recorded as such in the data. An explanation has been added that through using these code lists, the identification of recorded cases is highly accurate, but cases not recorded in the data sources remain undetectable (line 227).
Comments 4: Considering the incidence of cases, an approximation of the power of the study to detect significant differences between potential risk factors and outcomes could be reported.
Response 4: Given the type of study, a traditional sample size or power calculation was not feasible. To improve statistical efficiency and limit biases, four controls per case are sampled in the case-cohort design, and this is done 1,000 times. The number of included cases of myocarditis and/or pericarditis drives the number of risk factors that can be selected in the variable selection procedure.
Comments 5: An explanation of the methodology for the control of potential confounding factors could be indicated for a better understanding of the study.
Response 5: This study is exploratory in nature and does not aim to find causal associations (line 230). As such, we investigate a wide range of potential risk factors, but do not control for any potential confounding specifically. Instead, by performing multivariable regression analyses, some potential confounders are by design considered in the analyses.
Comments 6: The protocol indicates that variables included in the multivariable models are based on statistical criteria without considering the potential confounding factors. This situation could be clarified.
Response 6: As outlined above, this study is exploratory in nature and therefore does not explicitly aim to control for all possible confounding variables, which would not be feasible. In the manuscript we explained the variable selection procedure of risk factors before inclusion in multivariable models and in Supplementary Table 5 we provide a more detailed description of the variable selection procedure.
Comments 7: The authors mention some limitations of the study. However, a potential selection bias could occur, regarding the severity of the cases.
Response 7: We cannot rule out selection bias as more severe cases may have been more likely to be included in the data due to these cases presenting to the hospital more often. Data availability does unfortunately not allow for controlling for this. We have added this potential selection bias as a limitation in the manuscript discussion (line 357).
Reviewer 3 Report
Comments and Suggestions for Authors
This is an interesting study proposal to assess the incidence of pericarditis following administration of the mRNA-1273 vaccine (the Moderna vaccine). Notably, the authors will examine a wide range of comorbidities and pharmacological treatments, as detailed in the supplemental data.
This reviewer has several considerations:
-Please consider a complementary approach evaluating the incidence rate of myocarditis after vaccination (and/or COVID-19 infection) in comparison to the incidence of myocarditis during the 3–5 years preceding the COVID-19 pandemic (2015–2019).
-Since most residents in the studied countries received the Astra and Pfizer vaccines, while others received recombinant vaccines if they experienced side effects after the mRNA vaccine, please discuss why these vaccines were excluded from the potential multivariate analysis.
-The authors appear to assume, in some parts of the article, that only cases of pericarditis occurring within one month of vaccination are related to it. However, they also describe that the cases of pericarditis with longer-term onset (90 days, 6 months, 12 months — line 125) will be considered. This reviewer suggests including a descriptive and quantitative analysis of the time interval between any vaccination and any subsequent pericarditis, and then discussing the potential association, rather than presuming a short-term temporal relationship.
-Consider including the cumulative number of vaccine doses (particularly in cases involving mixed vaccine types) as a distinct variable when analyzing pericarditis incidence.
- Section 3.7.4: Alongside truncating the analysis post mRNA vaccine exposure, incorporate a long-term assessment spanning the entire period from vaccination to the present.
- Clarify the definition of ‘unexposed’: Does this refer to individuals vaccinated with other vaccine types or entirely unvaccinated individuals?
-Please also consider the potential selection bias related to polypharmacy, comorbidity, and vaccination, as it is likely that the most comorbid patients are those who choose to receive the vaccine, as some authors have previously suggested.
While the article is very relevant and well presented, these issues should be addressed to enhance its clarity and robustness.
Author Response
Comments 1: Please consider a complementary approach evaluating the incidence rate of myocarditis after vaccination (and/or COVID-19 infection) in comparison to the incidence of myocarditis during the 3–5 years preceding the COVID-19 pandemic (2015–2019).
Response 1: We thank the reviewer for this suggestion. However, incidence estimates were considered beyond the scope of this specific post-authorization safety study and have therefore not been performed.
Comments 2: Since most residents in the studied countries received the Astra and Pfizer vaccines, while others received recombinant vaccines if they experienced side effects after the mRNA vaccine, please discuss why these vaccines were excluded from the potential multivariate analysis.
Response 2: This study was registered as an EUPAS study and specifically designed to investigate the myocarditis/pericarditis safety signal following mRNA-1273 vaccination. Indeed, other vaccinations have also been associated with these outcomes, but those associations are beyond the scope of this study. In addition, individuals with myocarditis and/or pericarditis following a different COVID-19 vaccine were explicitly excluded from the study cohort, as we were interested in the comparison between post-mRNA-1273 myocarditis/pericarditis and myocarditis/pericarditis not associated with any COVID-19 vaccination (line 194). We agree that more research into the risk of myocarditis and/or pericarditis following those other vaccines is still warranted.
Comments 3: The authors appear to assume, in some parts of the article, that only cases of pericarditis occurring within one month of vaccination are related to it. However, they also describe that the cases of pericarditis with longer-term onset (90 days, 6 months, 12 months — line 125) will be considered. This reviewer suggests including a descriptive and quantitative analysis of the time interval between any vaccination and any subsequent pericarditis, and then discussing the potential association, rather than presuming a short-term temporal relationship.
Response 3: We thank the reviewer for this suggestion. However, the longer-term time periods outlined in line 125 do not relate to the study of the onset of post-vaccination pericarditis, but rather to the study of outcomes following post-vaccination myocarditis and/or pericarditis. The authors have decided to study the onset of myocarditis and/or pericarditis within 30 days following mRNA-1273 vaccination, as most existing literature suggests that these outcomes tend to occur shortly after vaccination (line 66). However, we are also interested in the clinical course of these outcomes, and how this clinical course compares to that of myocarditis and/or pericarditis not following mRNA-1273 vaccination. This is done in a separate cohort study design. We have added some wording to clarify the distinction between both study designs (Line 129).
Comments 4: Consider including the cumulative number of vaccine doses (particularly in cases involving mixed vaccine types) as a distinct variable when analyzing pericarditis incidence.
Response 4: We thank the reviewer for this suggestion and are happy to share that this is a variable of interest in the study, which was considered part of the description of vaccination characteristics. Some wording has been added (line 267 and line 282) to clarify that the number of doses received is considered in the study.
Comments 5: Section 3.7.4: Alongside truncating the analysis post mRNA vaccine exposure, incorporate a long-term assessment spanning the entire period from vaccination to the present.
Response 5: We appreciate the reviewer’s suggestion about the exposure window. However, based on existing evidence on the onset of myocarditis and/or pericarditis following COVID-19 vaccination, which suggests these events occur shortly after vaccination (see references 7-11), the authors have decided to not further extend the assessment window.
Comments 6: Clarify the definition of ‘unexposed’: Does this refer to individuals vaccinated with other vaccine types or entirely unvaccinated individuals?
Response 6: We have added a more detailed definition of unexposed cases to section 3.1 (Line 126).
Comments 7: Please also consider the potential selection bias related to polypharmacy, comorbidity, and vaccination, as it is likely that the most comorbid patients are those who choose to receive the vaccine, as some authors have previously suggested.
Response 7: We thank the reviewer for this suggestion and agree that some selection bias may occur in this study. A short discussion of potential selection biases has been added to the discussion of the manuscript (line 357).
Reviewer 4 Report
Comments and Suggestions for Authors
Authors need to present the measurement of the variables.
Moreover, authors need to present how the sample is adequate to accomplish the research objective.
Authors need to present the statistical outcomes using Tables.
Authors just presented it without concrete numbers.
Authors need to present threshold for the statistics.
Authors need to present academic value of this work more.
Author Response
Comments 1: Authors need to present the measurement of the variables.
Response 1: The measurement of variables has been clarified in Table S1. This is now also mentioned in the manuscript in line 240.
Comments 2: Moreover, authors need to present how the sample is adequate to accomplish the research objective.
Response 2: The study uses broadly representative data from five data sources from four European countries, covering millions of actively contributing participants. This representative European population allows the authors to address the research objectives in an adequate manner. Some clarification about the study population has been added in line 145.
Comments 3: Authors need to present the statistical outcomes using Tables. Authors just presented it without concrete numbers.
Response 3: The current manuscript is a protocol, hence no results are presented in this manuscript. Therefore, these can also not be presented in a table and no concrete numbers are available yet.
Comments 4: Authors need to present threshold for the statistics.
Response 4: A sentence explaining the statistical threshold used for interpreting associations has been added (line 274).
Comments 5: Authors need to present academic value of this work more.
Response 5: The authors have presented the value of this work in the discussion. One of the main contributions of this work is that it is one of the first studies to investigate risk factors and clinical course of myocarditis and pericarditis in so much detail, which we hope will serve as an inspiration for other researchers (line 339).
Round 2
Reviewer 2 Report
Comments and Suggestions for Authors
The authors have tried to address the points of our previous review. However, the weakness of this type of research remains. More active research might be more appropriate than research based on data collected by healthcare systems. Considering that this is an initial approach, the manuscript could be useful to found out a first approximation to myocarditis and pericarditis incidence related to mRNA-1273 vaccination.
Author Response
Comments 1: The authors have tried to address the points of our previous review. However, the weakness of this type of research remains. More active research might be more appropriate than research based on data collected by healthcare systems. Considering that this is an initial approach, the manuscript could be useful to found out a first approximation to myocarditis and pericarditis incidence related to mRNA-1273 vaccination.
Response 1:
We thank the reviewer for their kind words and their suggestion. Indeed, there are several inherent weaknesses to the use of real-world data for studying outcomes as post-vaccination myocarditis and pericarditis. At the same time, data on longer-term outcomes of patients who developed myocarditis or pericarditis during the pandemic has only become available at a later time point, and could not be studied more actively before. The limitations to the use of secondary health data have been mentioned in the discussion (line 349). Moreover, we agree with the reviewer that this study (for which this manuscript is the study protocol) will provide estimation in the incidence of myocarditis and pericarditis following mRNA-1273 vaccination.
Reviewer 3 Report
Comments and Suggestions for Authors
Dear editors,
Thank you for the opportunity to revise again a new version of this manuscript.
Dear authors,
Some of the questions posed previously are still not solved with complementary new data. Alternatively, enhanced discussion should consider different aspects:
If it’s not possible to compare the incidence of myocarditis after vaccination with the incidence rates in that concrete area in previous years, you are invited to compare myocarditis incidence rates with pre-existing local public health data or at least please contextualize your findings within the myocarditis incidence according to broader literature. Currently, this discussion is absent.
Since the study has not compared post-vaccination myocarditis rates with those following other vaccines, it would strengthen the analysis to specify the percentage of the study population receiving only the assessed vaccine and the rationale for excluding other vaccines from the comparison.
Given that the title indicates a study of risk factors for myocarditis and pericarditis following vaccination, focusing solely on 30-day outcomes may not fully align with this scope. The authors might consider either adjusting the title to specify short-term effects, or extending the observation period.
Additionally, they should explicitly acknowledge that long-term effects will not be assessed for the moment, and to clearly state the longest follow-up periods of the existing literature.
Finally, rather than general wording, we recommend presenting concrete data illustrating the relationship between myocarditis risk and the number of vaccine doses received.
Thank you for your work in this interesting field.
Author Response
Comments 1: If it’s not possible to compare the incidence of myocarditis after vaccination with the incidence rates in that concrete area in previous years, you are invited to compare myocarditis incidence rates with pre-existing local public health data or at least please contextualize your findings within the myocarditis incidence according to broader literature. Currently, this discussion is absent.
Response 1: In this protocol, we list some previous incidence estimates from existing literature (e.g., lines 66-71). As this is a protocol, we do not yet have findings and are therefore unable to discuss the findings in light of pre-existing incidence estimates. In addition, the study has several specific objectives, which do not include incidence estimation. However, we do plan to compare the incidence of myocarditis and pericarditis that we find during the study with existing estimates to assess if numbers are as expected. We will consider the reviewer’s suggestion on contextualizing findings according to broader literature when results become available.
Comments 2: Since the study has not compared post-vaccination myocarditis rates with those following other vaccines, it would strengthen the analysis to specify the percentage of the study population receiving only the assessed vaccine and the rationale for excluding other vaccines from the comparison.
Response 2: We thank the reviewer for this suggestion. However, it is not possible to explicitly report the percentages of the study population receiving the mRNA-1273 vaccine because this is a study protocol. We do plan to report this with the study results once available. The rationale for not studying other vaccines is that this is a registered real-world data study explicitly aimed at studying the mRNA-1273 vaccine (line 109). Additionally, in the cohort study design, recipients of other COVID-19 vaccines are explicitly excluded to allow for a comparison between post-mRNA-1273 cases and cases that are unrelated to vaccination, which is explained in line 197.
Comments 3: Given that the title indicates a study of risk factors for myocarditis and pericarditis following vaccination, focusing solely on 30-day outcomes may not fully align with this scope. The authors might consider either adjusting the title to specify short-term effects, or extending the observation period.
Response 3: We appreciate the comment by the reviewer, but changing the title by including 30-days outcome or short-term effects (because of the case-cohort analysis part) would wrongfully reflect the entire scope of the study, because the clinical course and outcomes which are part of the cohort analysis have a follow-up up to 12 months. Therefore, to ensure correctness and clarity of the study title, we strongly prefer to keep the title as it is.
Comments 4: Additionally, they should explicitly acknowledge that long-term effects will not be assessed for the moment, and to clearly state the longest follow-up periods of the existing literature.
Response 4: We have incorporated a more extensive rationale for defining post-vaccination myocarditis and pericarditis to occur within 30 days following vaccination, including references to existing literature (line 119). Moreover, please refer to the response to comment 3, which emphasizes that long-term outcomes (up to 12 months) will be studied in the cohort analysis.
Comments 5: Finally, rather than general wording, we recommend presenting concrete data illustrating the relationship between myocarditis risk and the number of vaccine doses received.
Response 5: We thank the reviewer for this suggestion and are indeed planning on presenting the suggested numbers. More precisely, the exact number of vaccination doses received will be reported with the study results, as highlighted in lines 270 and 285. The current manuscript, however, is a study protocol, and these numbers are therefore not presented, as results are not yet available.
Round 3
Reviewer 3 Report
Comments and Suggestions for Authors
Thank you for the opportunity to review this revised version.
I still believe that the numbers they report for incidence of myocarditis following different types of mRNA vaccines should be compared with preexisting data of such incidence in the introductions. Since authors plan to compare their findings with that data, simply state it in the discussion and not only in the response letter.
Again, if authors plan to show the percentages of the study population receiving the mRNA-1273 vaccine, state in the text that they do plan to report this with the study results once available.
Authors should clearly indicate in the text why controls will be followed during 12 months and why the myocarditis group will only be considered to be due to the vaccine if it occurs during the first 30 days, instead of performing a comparison between both groups at the end on an equal follow-up period (I strongly recommend to perform both analysis). The short follow-up period of the previous literature is not a reason for missing a longer follow up period now.
I look forward to know the results of this study once performed.

Author Response
Comments 1: I still believe that the numbers they report for incidence of myocarditis following different types of mRNA vaccines should be compared with preexisting data of such incidence in the introductions. Since authors plan to compare their findings with that data, simply state it in the discussion and not only in the response letter.
Response 1: We understand the reviewer’s suggestion, and have now explicitly stated that incidences will be compared in line 232.
Comments 2: Again, if authors plan to show the percentages of the study population receiving the mRNA-1273 vaccine, state in the text that they do plan to report this with the study results once available.
Response 2: We thank the reviewer for this suggestion and have added some wording to explicitly state this in line 272.
Comments 3: Authors should clearly indicate in the text why controls will be followed during 12 months and why the myocarditis group will only be considered to be due to the vaccine if it occurs during the first 30 days, instead of performing a comparison between both groups at the end on an equal follow-up period (I strongly recommend to perform both analysis). The short follow-up period of the previous literature is not a reason for missing a longer follow up period now.
Response 3: We would like to point out to the reviewer that controls in the case-cohort analysis of this study are the individuals who did not develop myocarditis and/or pericarditis after mRNA-1273 vaccination, and these people are not followed up for 12 months as the reviewer indicated. The 12-month follow-up only concerns the cohort analysis, in which myocarditis and pericarditis cases regardless of their underlying etiology (i.e., vaccine-related or not) are followed up. The two main groups of interest are myocarditis/pericarditis cases that occurred within 30 days following vaccination and cases that were not associated with vaccination. We understand the reviewer’s point that myocarditis/pericarditis may (incidentally) occur more than 30 days after vaccination. However, the longer the time period after vaccination, the more uncertain the presumable causal relationship between vaccination and myocarditis/pericarditis event becomes. E.g., especially during the pandemic individuals could often get infected by SARS-CoV-2, which is known to be a stronger risk factor for myocarditis/pericarditis development, and a longer follow-up would increase the likelihood of including post-infection instead of post-vaccination cases. Nevertheless, we have mentioned the inclusion criterium as a limitation in the discussion section (line 364-366).